# Quantification of Liver Fat Content with the iATT Algorithm: Correlation with Controlled Attenuation Parameter

**DOI:** 10.3390/diagnostics12081787

**Published:** 2022-07-23

**Authors:** Giovanna Ferraioli, Ambra Raimondi, Laura Maiocchi, Annalisa De Silvestri, Carlo Filice

**Affiliations:** 1Dipartimento di Scienze Clinico-Chirurgiche, Diagnostiche e Pediatriche, Università di Pavia, Viale Brambilla 74, 27100 Pavia, Italy; ambra.raimondi01@universitadipavia.it (A.R.); carfil@unipv.it (C.F.); 2Dipartimento di Scienze Mediche e Malattie Infettive, Fondazione IRCCS Policlinico San Matteo, Viale Camillo Golgi 19, 27100 Pavia, Italy; l.maiocchi@smatteo.pv.it; 3Clinical Epidemiology and Biometric Unit, Fondazione IRCCS Policlinico San Matteo, Viale Camillo Golgi 19, 27100 Pavia, Italy; a.desilvestri@smatteo.pv.it

**Keywords:** liver steatosis, fat quantification, attenuation coefficient, fatty liver, ultrasound, controlled attenuation parameter, NAFLD, chronic liver disease, concordance studies

## Abstract

Background: The primary aim of our study was to assess the correlation between an improved version of the attenuation coefficient available on the Arietta 850 ultrasound system (iATT, Fujifilm Healthcare, Tokyo, Japan) and controlled attenuation parameter (CAP). The secondary aim was to assess whether focusing only on iATT acquisition without following the strict protocol for liver stiffness measurements would affect iATT measurement. Methods: Consecutive individuals were enrolled. Pearson’s *r* was used to test the correlation between ATT and CAP values. The concordance between iATT and CAP was tested using Lin’s concordance correlation coefficient (CCC). Results: 354 individuals (203 males, 151 females) were studied. The overall Pearson correlation between CAP and iATT values obtained following or not following the liver stiffness measurement protocol, respectively, were *r =* 0.73 and *r =* 0.71. The correlation was affected by the interquartile range/median (IQR/M) of the 10 measurements: it was *r =* 0.75 for IQR/M ≤ 15% and *r =* 0.60 for IQR/M > 15%. CCC showed that there was a moderate to good concordance between iATT and CAP values. Conclusion: iATT shows a strong correlation with CAP that does not decrease when the protocol for liver stiffness acquisition is not followed. The correlation between iATT and CAP values is higher when the IQR/M ≤ 15%.

## 1. Introduction

Non-alcoholic fatty live disease (NAFLD) has become the most common chronic liver disease worldwide, with a prevalence in the adult population estimated at around 25% in 2015 and projected at 33.5% in 2030, an increase that parallels the epidemics of obesity, diabetes and metabolic syndrome [1,2]. NAFLD is an umbrella term that includes a spectrum of chronic liver disease, from simple steatosis (NAFL) to non-alcoholic steatohepatitis (NASH) that may lead to fibrosis, and eventually cirrhosis with its complications. It has been reported that about 20% of NAFLD patients will develop NASH [2]. Therefore, a vast majority of patients have simple steatosis that generally does not require a histologic assessment.

However, a recent study in a nationwide adult cohort has reported an overall mortality significantly higher in all of the NAFLD patients, including those with simple steatosis, than in controls [3]. The liver fat content is linked to the metabolic syndrome and the cardiovascular risk [4]. Moreover, significant steatosis is associated with fibrosis progression in NAFLD patients [5]. Therefore, a noninvasive method to quantify liver fat content is of great interest.

The controlled attenuation parameter (CAP), which measures the attenuation of the ultrasound beam and is obtained together with the liver stiffness measurement using the FibroScan device (Echosens, Paris, France), is a tool that has been available for more than a decade, and several studies have assessed its value in the quantification of liver fat content [6]. 

In the last few years, several ultrasound (US) manufacturers have developed software for quantifying liver fat content by means of the attenuation coefficient [7,8,9]. Among them, iATT is the upgraded version of the ATT algorithm released by Fujifilm Healthcare, Tokyo, Japan (previously Hitachi Ltd., Tokyo, Japan) a few years ago. As with CAP, the attenuation coefficient is obtained together with the liver stiffness measurement, the iATT measurement area is not user-adjustable, and the assessment is made in a fixed area that is not color-coded. The manufacturer reports that, in the improved algorithm, the factors that caused signal deterioration in the far field have been eliminated and the calculation method has been modified. Moreover, the length of the measurement area has been narrowed and set at a distance of 35 mm from the skin in the near field to 75 mm in the far field, and two horizontal lines graphically show the width and the length of the measurement area (Figure 1 and Figure 2).

By using the previous version of the algorithm, a study reported a moderate correlation with CAP (*r =* 0.55 for values obtained with the M probe of the Fibroscan and *r =* 0.53 for values obtained with the XL probe) [10]. 

The primary aim of our study was to assess the correlation between iATT and CAP values in a series of consecutive patients. The secondary aim was to assess whether focusing only on the iATT acquisition without following the strict protocol for liver stiffness measurements would affect the reproducibility of the iATT values.

## 2. Materials and Methods

This was a cross-sectional study. Between June 2021 and April 2022, consecutive adult individuals referred to the ultrasound unit of the department of medical sciences and infectious diseases of our institution for fat quantification with the CAP, and who voluntarily also accepted to undergo assessment with the iATT algorithm, were prospectively enrolled. The subject’s characteristics and bio-humoral tests, when available, were recorded. All of the individuals had an alcohol intake less than 20 g/day. The exclusion criteria were failure or unreliable results with the vibration controlled transient elastography (VCTE). 

### 2.1. iATT Measurements

The examinations were performed by using the Arietta 850 US system (Fujifilm Healthcare, Tokyo, Japan) with a convex broadband probe. Steatosis with B-mode ultrasound was scored as follows: absent (score 0) when there was a normal liver echotexture; mild (score 1) steatosis, in case of a slight and diffuse increase in fine parenchymal echoes with normal visualization of diaphragm and portal vein borders; moderate (score 2) steatosis, in case of a moderate and diffuse increase in fine echoes with slightly impaired visualization of portal vein borders and diaphragm; severe (score 3) steatosis, in case of marked increase of fine echoes with poor or no visualization of portal vein borders, diaphragm, and posterior portion of the right liver lobe [9]. Since the attenuation coefficient is obtained together with liver stiffness, the protocol required for accurate liver stiffness values was followed [11,12,13]. Ten valid measurements were obtained and the median value in kiloPascal (kPa) for stiffness and in decibel/meter/megahertz (dB/m/MHz) for fat quantification, as well as the interquartile range/median (IQR/M) ratios, were used for statistical analysis. The validity of each measurement was assessed using the manufacturer’s quality criterion, i.e., a VsN ≥ 50% [14]. If this criterion was not fulfilled, the examination was considered a failure. This set of measurements took up to 4 minutes.

Another set of 10 “only iATT” measurements were obtained. Since the size of ROI for the stiffness measurement is smaller than that for fat quantification, this time we focused only on the best image for iATT, i.e., the homogenous liver parenchyma without any vessel inside the measurement area of the attenuation coefficient and without taking into account the VsN (Figure 3). This set of measurement took less than one minute in all of the individuals. The median value in dB/m/MHz and the IQR/M were used for statistical analysis.

In all of the individuals, the skin-to-liver capsule distance was measured.

### 2.2. Controlled Attenuation Parameter Measurements

The CAP was obtained by using the FibroScan 502 Touch system. The 3.5 MHz M probe was used when the skin to liver capsule distance, estimated with ultrasound, was ≤ 25 mm, otherwise the 2.5 MHz XL probe was used. The FibroScan estimates both the liver stiffness in kPa and the liver attenuation coefficient in decibel/meter (dB/m). The principles of CAP have been described elsewhere [15]. The FibroScan computes CAP only when the associated liver stiffness measurement (LSM) is valid, and it uses the same signal as the one used to measure liver stiffness. As recommended, only the LSM with 10 validated measurements and an IQR/M ≤ 30% were considered reliable [11]. There are no recommendations for successful CAP measurement.

All of the subjects gave their informed consent for inclusion before they participated in the study. The study was conducted in accordance with the Declaration of Helsinki, and the protocol was approved by the Ethics Committee of Fondazione IRCCS Policlinico San Matteo (P-20170022247).

### 2.3. Statistical Analysis

The descriptive statistics were produced for the demographic characteristics for this study sample of patients. The Shapiro–Wilk test was used to test the normal distribution of quantitative variables. When quantitative variables were normally distributed, the results were expressed as the mean value and standard deviation (SD), otherwise the median and the IQR (25th–75th percentile) were reported. The qualitative variables were summarized as counts and percentages. Pearson’s *r* was used to test the correlation between iATT and CAP values. The concordance between iATT and CAP was tested using Lin’s concordance correlation coefficient (CCC) [16]. To assess the concordance between the two methods, the iATT measurement was converted in the same unit of measure of CAP multiplying the value by 100 (centimeter to meter) and by 3.5, which is the ultrasound frequency at which the CAP is computed, regardless of the ultrasound frequency of the probe being used. 

CCC and Pearson’s *r* range in values from 0 to +1. The Bland and Altman limits of agreement (LOA), with their 95% confidence interval (CI), were also calculated. The correlation and concordance were classified as poor (0.00 to 0.20), fair (0.21 to 0.40), moderate (0.41 to 0.60), good (0.61 to 0.80), or excellent (0.81 to 1.00) [17].

*p* < 0.05 was considered statistically significant. All of the tests were two-sided. The data analysis was performed with the STATA statistical package (release 17.0, 2021, Stata Corporation, College Station, TX, USA).

## 3. Results

Three hundred and fifty-four patients (203 males and 151 females) were studied. One hundred and forty-six patients were affected by chronic hepatitis C; 80 patients by NAFLD; 48 patients by chronic hepatitis B; the remaining patients had other etiologies of liver disease. The main clinical and demographic characteristics of the study cohort are reported in Table 1. The XL probe was used in 46 (13%) of the patients because the skin-to-liver capsule distance was higher than 25 mm. 

Failures with liver stiffness measurement and iATT were observed in 74 (20.9%) of the cases. The main determinant of the failures was the skin-to-liver capsule distance (odds ratio: 25.5; 95% confidence interval: 11.6–56.1). 

### Correlation and Concordance Analysis

The overall Pearson correlation between CAP and iATT values obtained following the liver stiffness measurement protocol was *r* = 0.73 (*p* = 10^−5^) (Figure 4), and it was *r =* 0.71 (*p* = 10^−5^) when the iATT measurement was obtained without following the stiffness protocol. 

The correlation was affected by the IQR/M of the 10 measurements: it was *r =* 0.75 (*p* = 10^−5^) in the 203 cases showing an IQR/M of the iATT value ≤ 15%, and the correlation was the same (*r =* 0.75, *p* = 10^−5^) in the 130 cases with an IQR/M ≤ 10%, whereas it was *r =* 0.60 (*p* = 10^−5^) in the 77 cases with an IQR/M >15% (Table 2).

Lin’s CCC analysis showed that there was a moderate to good concordance between the iATT and CAP values (Table 2). 

The iATT values obtained without following the strict protocol for liver stiffness measurements were obtained in 220 patients of this cohort, and there was no statistically significant difference with the remaining patients of the study cohort. The Pearson correlation was *r* = 0.71 (*p* = 10^−5^).

## 4. Discussion

The results of this study show a good correlation between the improved iATT algorithm for the assessment of the attenuation coefficient and CAP, with a progressive and linear increase in the values up to the grade of severe liver steatosis. This correlation is higher than that reported in a published study, which was performed with a previous version of the algorithm [10], and seems to indicate that iATT is a promising tool in the evaluation of patients with suspected fatty liver. The correlation was not affected by changing the acquisition plane and without following the recommended protocol for stiffness measurement. It must be highlighted that, in this latter case, the time required for a set of 10 measurement was less than one minute. This result shows that the attenuation coefficient is easier to obtain in relation to the liver stiffness measurement and seems to indicate that the strict protocol that must be followed for liver stiffness measurements is not required. 

It is worth noting that the attenuation coefficient with iATT is obtained in a fixed area and the depth of the measurement cannot be changed. It is likely that there is a depth dependence of the attenuation coefficient measurement that should be assessed in future studies, as recently recommended by the American Institute of Ultrasound in Medicine (AIUM)-RSNA Quantitative Imaging Biomarkers Alliance (QIBA) Pulse-Echo Quantitative Ultrasound (PEQUS) initiative for fat quantification [8]. 

Our data show that the strength and direction of the linear relationship between iATT and CAP was good, however the agreement between the measurements was only moderate. This study, which was specifically designed to compare the results obtained with iATT to those obtained with CAP in order to assess the correlation, did not evaluate the performance of the upgraded algorithm using CAP as reference standard. 

This study simply assessed the relationship between the two variables, i.e., iATT and CAP. No gold standard was used and both of these algorithms may not be optimal in quantifying liver fat content. CAP has been available for more than a decade and has become a point-of-care technique due in part to its easy use. However, the performance of CAP seems suboptimal and the cutoffs for detecting and grading liver steatosis vary considerably between studies, ranging from 219 dB/m in a cohort of patients with chronic hepatitis C to 294 dB/m in a metanalysis of NAFLD patients [18,19]. It must be underscored that the prevalence of the disease, i.e., the presence of fatty liver in the studied populations, rather than the etiology of liver disease might likely explains this variability [20]. Both the recent position paper of the World Federation for Ultrasound in Medicine and Biology on liver fat quantification and the AIUM-RSNA QIBA initiative article have highlighted that CAP is not an adequate reference standard for evaluating the accuracy of the emerging ultrasound techniques for fat quantification with attenuation coefficients [7,8]. 

There are some limitations to this study. First, we did not assess the performance of iATT. Second, we did not compare the results obtained with iATT with those of other algorithms by other vendors that are also based on the attenuation coefficient. Third, the agreement between the readers was not assessed. 

In conclusion, iATT shows a strong correlation with CAP that seems not to be decreased when the protocol for liver stiffness acquisition is not followed. The correlation of iATT with CAP values seems affected by the variability between measurements and it is higher when the IQR/M ≤ 15%.

## Figures and Tables

**Figure 1 diagnostics-12-01787-f001:**
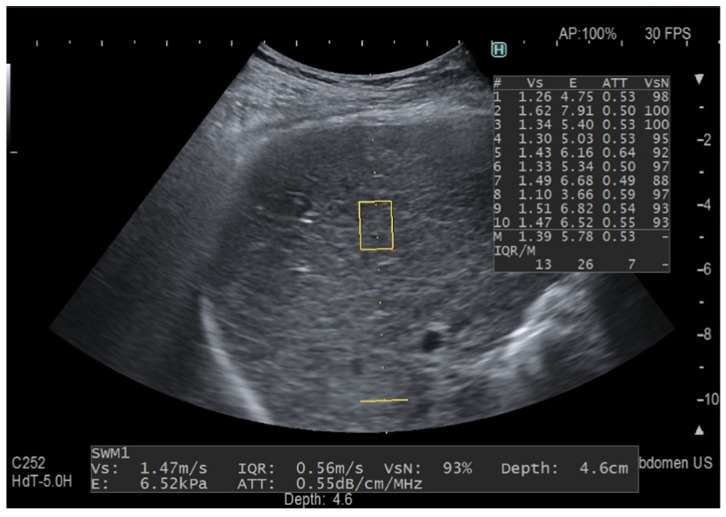
Fat quantification with the previous ATT algorithm. The yellow rectangle is the region of interest (ROI) for stiffness measurement and the yellow horizontal line indicates the depth of the ATT measurement area in the far field, which is set at 100 mm. The ATT measurement area has a fixed size and the measurement of the attenuation coefficient is given in dB/cm/MHz together with liver stiffness measurement, which is shown both in m/s and kPa. ATT quantifies liver fat content in an areathat has a length of 6 cm and is set at 40–100 mm from the skin. This measurement was taken in a 64-year-old patient with chronic hepatitis C following the protocol for liver stiffness measurement and with a VsN always ≥50%. The attenuation coefficient value is within the normal range.

**Figure 2 diagnostics-12-01787-f002:**
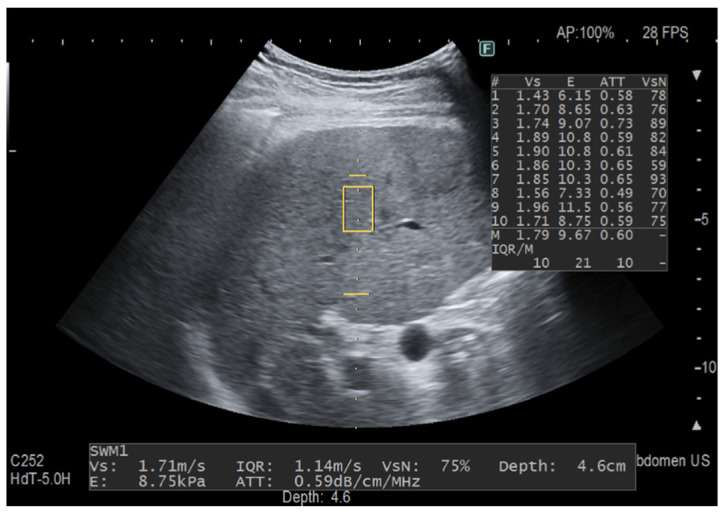
Fat quantification with iATT. The two horizontal yellow lines graphically show the width (length of each line) and the length (distance between the two lines) of the iATT measurement area, which has a fixed size (length of 4 cm, from 35 to 75 mm from the skin). The measurement of the attenuation coefficient is given in dB/cm/MHz together with liver stiffness measurement. The yellow rectangle is the region of interest (ROI) for stiffness measurement. This measurement was taken in a 58-year-old patient with primary biliary cirrhosis but not steatosis.

**Figure 3 diagnostics-12-01787-f003:**
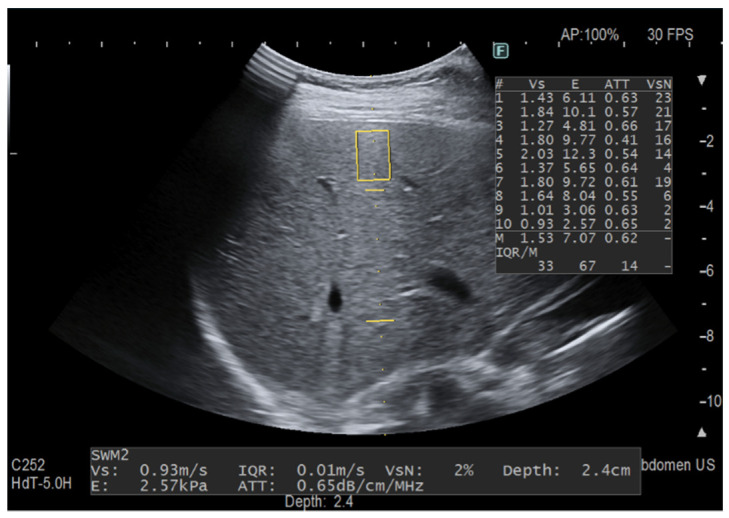
The measurement was taken focusing on the best image for iATT without following the protocol for stiffness assessment. The two horizontal yellow lines graphically show the width (length of each line) and the length (distance between the two lines) of the iATT measurement area, which has a fixed size (length of 4 cm, from 35 to 75 mm from the skin). The yellow rectangle is the region of interest (ROI) for stiffness measurement. Because it is not possible to exclude the stiffness measurement, the stiffness ROI was intentionally positioned close to the liver capsule. This explains the huge variability between consecutive stiffness measurements with an IQR/M = 67% and a VsN always <50%.

**Figure 4 diagnostics-12-01787-f004:**
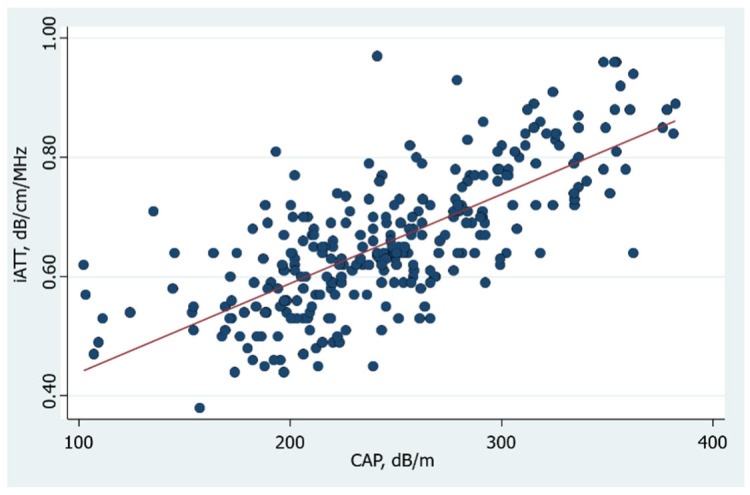
Graph of the overall correlation of iATT with CAP.

**Table 1 diagnostics-12-01787-t001:** Characteristics of the study cohort (*n* = 354).

Gender, men (%)	203 (57.3)
Age, years (SD)	59.7 (14.5)
NAFLD, *n* (%)	80 (22.6)
Hepatitis C, *n* (%)	146 (41.2)
Hepatitis B, *n* (%)	48 (13.6)
Other etiologies of liver disease, *n* (%)	80 (22.6)
BMI, kg/m^2^ (SD)	26.8 (4.7)
AST, IU/L (IQR)	26 (20–46)
ALT, IU/L (IQR)	30.5 (18–55.5)
GGT, IU/L (IQR)	38.5 (21–81)
Platelets count, 10^3^/mm^3^, (IQR)	211 (171–258)
Stiffness value, SWM, kPa (IQR)	5.87 (4.34–8.39)
Stiffness value, VCTE, kPa (IQR)	6.1 (4.6–8.4)
Steatosis B-mode US score 0 (%)	189 (53.5)
Steatosis B-mode US score 1 (%)	66 (18.7)
Steatosis B-mode US score 2 (%)	63 (17.9)
Steatosis B-mode US score 3 (%)	39 (9.9)
CAP, dB/m (IQR)	251 (210–299)
iATT, dB/cm/MHz (IQR)	0.64 (0.58–0.72)

SD: standard deviation; IQR: interquartile range; BMI: body mass index; AST: aspartate aminotransferase; ALT: alanine aminotransferase; GGT: gamma-glutamyl transferase; SWM: shear wave measurement (ultrasound system); VCTE: vibration controlled transient elastography (FibroScan system); US: ultrasound; CAP: controlled attenuation parameter, iATT (attenuation, ultrasound system).

**Table 2 diagnostics-12-01787-t002:** Concordance between ATT and CAP values.

Variable	*N*=	CCC (95% CI)	Pearson’s *r*	Mean Difference, dB/m (95% Limits of Agreement)
Overall	280	0.66 (0.60–0.71)	0.73	−16 (−92–60)
BMI ≤ 25	136	0.61 (0.51–0.70)	0.67	−12 (−79–56)
BMI 25.1–29.9	98	0.56 (0.44–0.67)	0.65	−16 (−102–70)
BMI ≥ 30	45	0.58 (0.41–0.74)	0.70	−28 (−100–43)
Skin-to-liver capsule distance ≤ 20 mm	195	0.58 (0.50–0.67)	0.63	−9 (−81–63)
Skin-to-liver capsule distance > 20 mm	85	0.51 (0.39–0.63)	0.66	−31 (−107–44)
IQR/M ≤ 10	130	0.66 (0.57–0.74)	0.75	−19 (−97–58)
IQR/M ≤ 15	203	0.67 (0.61–0.74)	0.75	−14 (−91–63)
IQR/M > 15	77	0.51 (0.37–0.64)	0.60	−20 (−91–51)

## Data Availability

The data presented in this study are available on request from the corresponding author. The data are not publicly available due to the privacy of the patients.

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
