# Peer review of "Quantification of Liver Fat Content with the iATT Algorithm: Correlation with Controlled Attenuation Parameter"

_diagnostics, 2022, doi:10.3390/diagnostics12081787_

Round 1

Reviewer 1 Report

This study showed a good correlation between the improved iATT 185 algorithm for the assessment of the attenuation coefficient and CAP, with a progressive 186 and linear increase of values up to the grade of severe liver steatosis and analyzed 354 patients.  This study was good design and protocol expansive research.  How about the relationship of CT or MRI image in severe liver steatosis.   I hope the authors should continue to do this research and have the analysis the relationship of CT or MRI image in severe liver steatosis.    

Author Response

Thank you very much for your comments. This study was designated only to evaluate whether there was any improvement in the correlation between the upgraded version of the iATT algorithm and the CAP that has become a point-of-care technique despite its several limitations. Therefore, it’s a study about correlation and not about performance.

Reviewer 2 Report

This is an interesting research to assess the correlation between CAP and iATT beyond ATT. However, there are some unsolved things in this research.

1.     Compared with previous research (Tamaki et al. Hepatol Res 2018; 48: 821-828), what is the novel findings about iATT beyond ATT? There was a moderate correlation of ATT with te fat area (r = 0.50) in previous study. Is it possible to briefly describe the strengths of iATT that differentiate it from ATT??

2.     As you aforementioned in the manuscript, CAP is not an accurate reference standard for evaluating the accuracy of the emerging ultrasound techniques for fat quantification using attenuation coefficients. Could you please explain why you used CAP as a comparison object?

3.     If iATT and CAP valuer are moderate to good concordance, is there a rough COV of iATT that is correlated with CAP?

4.     Is this method (IATT) effective in patients with having ascites G1 or 2 ?

5.     How about inserting a graph showing the linear relationship between iATT and CAP?

6.     There is no difference in CCC between the two groups at the skin-to-liver capsule distance based on 2 cm. How should this be interpreted?

Thank you for your nice research.

Author Response

  1. Compared with previous research (Tamaki et al. Hepatol Res 2018; 48: 821-828), what is the novel findings about iATT beyond ATT? There was a moderate correlation of ATT with te fat area (r = 0.50) in previous study. Is it possible to briefly describe the strengths of iATT that differentiate it from ATT??

This study was designated only to evaluate whether there was any improvement in the correlation between the upgraded algorithm and CAP, which has become a point-of-care technique despite its several limitations. Therefore, it’s a study about correlation, and not about performance, and the correlation is much better than that of the previous algorithm. It has been detailed in the introduction and discussion of the submitted manuscript.

  1. As you aforementioned in the manuscript, CAP is not an accurate reference standard for evaluating the accuracy of the emerging ultrasound techniques for fat quantification using attenuation coefficients. Could you please explain why you used CAP as a comparison object?

Despite its several limitations, CAP is considered a point-of-care technique for the quantification of liver fat content. Therefore, we aimed at assessing the correlation of CAP with what is considered the available standard for steatosis assessment. It has been detailed in the introduction and the discussion of the submitted manuscript.

  1. If iATT and CAP valuer are moderate to good concordance, is there a rough COV of iATT that is correlated with CAP?

The median COV of iATT was 9.9 (IQR 7.2-12.9) and it was not correlated with CAP (r=-0.10).

  1. Is this method (IATT) effective in patients with having ascites G1 or 2 ?

Yes, it is. This method measures the attenuation of the ultrasound beam as it traverses tissue.

  1. How about inserting a graph showing the linear relationship between iATT and CAP?

Thank you very much for this comment. The graph has been inserted.

  1. There is no difference in CCC between the two groups at the skin-to-liver capsule distance based on 2 cm. How should this be interpreted?

Both iATT and CAP measure the attenuation of the ultrasound beam and both are affected by the skin-to-liver capsule distance. Therefore, it’s not a surprise that there wasn’t any difference in CCC between the two algorithms.

Round 2

Reviewer 2 Report

I think the authors submitted meticulous responses and corrections.

Thank you for getting it resolved my requirement.  I don't have any additional comments or requests.

Thank you for your meticulous responses and corrections. 

Author Response

Thank you!